# OpenReview forum: "AlgoVeri: An Aligned Benchmark for Verified Code Generation on Classical Algorithms"
_ICML.cc/2026/Conference — ICML 2026 spotlight_

### Official Review · Reviewer_QDYe · 2026-03-09

**Soundness:** 4
**Presentation:** 3
**Significance:** 3
**Originality:** 3
**Overall Recommendation:** 4
**Confidence:** 2

**Summary:**

The paper presents a benchmark for evaluating the capability of large language models for generation of formally verified code, given specifications. It focuses on Dafny, Verus and Lean. An in-depth performance analysis of open-source and frontier models reveals differences in their capabilities both among the chosen languages and the models themselves.

**Compliance With Llm Reviewing Policy:**

Affirmed.

**Final Justification:**

Systematically evaluating the ability of large language models to generate formally verified code is crucial for assessing their reliability in safety-critical software development, making the paper well motivated and a significant contribution to both the software verification and machine learning communities. The technical soundness of the technical claims and the experimental evaluation is excellent. The presentation needs some improvements, but the rebuttal acknowledges this fact and I believe that corresponding edits before publication are straightforward.

The limited size of the benchmark suite remains the main weakness of the submission. The rebuttal acknowledges this point, and I am convinced that a more in-depth discussion of this limitation will improve the paper. Nonetheless, the benchmark suite will remain limited for now.

Overall, the rebuttal did not change my impression of the paper strong enough to raise (or lower) my score. I still believe that the paper's strengths outweigh its weaknesses and lean towards acceptance.

**Key Questions For Authors:**

1. How sensitive is the benchmark to language updates?
2. Could the LLMs (maybe intentionally) create a solution which tricks the LLM-judge (either false positive or false negative)?

**Limitations:**

The authors do not discuss limitations of their investigation. However, I do not foresee any negative societal impact of this work.

**Strengths And Weaknesses:**

# Strengths
- The paper addresses important questions considering current LLM usage for programming.
- The presentation is very detailed.
- The findings from the analysis provide valuable insight going forward.

# Weaknesses
- The set of problems is small.
- The presentation could sometimes be more specific.

# Detailed comments
Due to the large and rising popularity of using LLMs for programming, the topic and contributions of this paper are timely and highly relevant. In particular, systematically evaluating the ability of large language models to generate formally verified code is crucial for assessing their reliability in safety-critical software development, making the paper well motivated and a valuable contribution to both the software verification and machine learning communities.

The paper is generally well written. In particular, Section 2 does a good job of introducing the relevant background for the work. However, the paper could benefit from assuming less prior knowledge on the part of the reader and providing clearer explanations of key concepts and terminology. For example, abbreviations such as SMT are used without ever introducing or spelling out the full term. Additionally, a few other aspects of the presentation should be improved as well:
- The confidence intervals in Table 2 are hard to read due to their font size. Moreover, given the relatively small number of problems in the dataset, their statistical significance and interpretability appear questionable.
- All lines in Figure 5 overlap, making it very hard to discern anything useful from the figure. Another diagram style (e.g., bar diagram) might be more appropriate in this situation.

Apart from the points above, the paper would also benefit from being more specific and rigorous. For instance, it is not entirely clear whether some of the comparisons the authors draw are fundamentally well-aligned (e.g., for example, between Dafny and Lean on the same problem): a task formalized in Lean could differ substantially in difficulty or required proof structure compared to Dafny, which raises the question of whether such comparisons are meaningfully comparable. Furthermore, several claims in Table 1 are difficult to verify without closely investigating each individual benchmark. For example, the statement that "unlike benchmarks derived from introductory coding problems, ALGOVERI targets complex algorithms requiring global invariants" uses terms such as "complex" that are not clearly defined. Similarly, the claim that “effective self-correction is an emergent capability” appears to go beyond what is directly supported by the presented evidence. Finally, the claim "Overall, these results indicate that current models have difficulty generalizing from local logic to reasoning about global properties" is not directly measured but only deduced from a combination of assumptions and proxy measurements.

The paper stands out in Sections 3 and 4, where it discusses the evaluation results. This is due both to the apparent difficulty of the benchmark and to the authors' effort to analyze different sources of failure in detail. Overall, these sections are very well executed and convincingly demonstrate that the benchmark provides meaningful insights. More specifically, the findings provided in Section 3 and to a larger extent in Section 4 give actionable feedback and show the value of the benchmark, Subsection 4.1 gives specific insight into differences between models, and Subsection 4.2 tries to give insight into the underlying mechanisms, shedding light on the differences of the ecosystems in the context of using LLMs.

The biggest issue of the paper is likely the limited size of the data set, making statements of statistical significance very challenging. In particular, low scores of all models may not be statistically meaningful without a very large problem set, and the small scale limits the benchmark in general. Additionally (and perhaps understandably), it is unfortunate that the recent jump in performance of coding agents, such as using Claude Opus 4.6 and Codex 5.3, is not investigated.

In conclusion, the paper addresses a relevant and timely topic. Its data set and insights appear insightful and helpful. Despite some issues with the presentation, I think the paper is available contribution and should be accepted.

---

> ### Author Rebuttal · Authors · 2026-03-30
>
> We thank Reviewer QDYe for the thorough and constructive feedback, and are encouraged that the reviewer found the benchmark valuable and Sections 3–4 insightful. We address the main concerns below and will revise the paper to improve specificity, presentation, and explicit discussion of limitations.
>
> >On benchmark size and newer models.
>
> We agree that the benchmark is modest in absolute size; we provide a more detailed discussion of this design tradeoff and expansion plan in our response to Reviewer oHS2.
>
> We added GPT-5.3-Codex under an 8-round refinement budget. Its compiler-verified rates are **49.35/14.29/23.38** on Dafny/Verus/Lean, and **42.86/11.69/11.69** after semantic filtering. These results do not qualitatively change the main picture: performance remains much stronger in Dafny than in Verus or Lean, and the semantic filter still removes a nontrivial fraction of outputs. Moreover, Codex still achieves **0% on graph problems**, reinforcing our conclusion that the benchmark’s hardest categories remain challenging even for newer coding agents.
>
> >On cross-language fairness and alignment.
>
> We do not claim that the verification tasks are equally difficult across languages. Indeed, the proof burden differs fundamentally between SMT-based and ITP-based systems, both because of toolchain-specific proof styles and because some formulations are more or less idiomatic in different ecosystems. What AlgoVeri controls is the **algorithmic property being verified**: we align the functional contracts and intended theorem complexity across Dafny, Verus, and Lean, so that differences in model performance are less confounded by differences in the underlying problem itself. Prior multilingual benchmarks often vary both the problem and the tool at the same time, making it difficult to disentangle whether a performance gap comes from problem mismatch or from the verification paradigm. AlgoVeri is designed to control the former so that the remaining differences are diagnostically informative about the interaction between models and verification systems. We agree that this point should be stated more carefully, and in the revision we will make the claim more explicit and more modest: the benchmark supports **controlled cross-paradigm comparison**, not an assertion that the tasks are equally easy in every language.
>
> >On claim specificity and wording.
>
> We agree that several claims in the current draft should be stated more carefully. In the revision, we will tighten the wording in three places. First, we will make “complex” more operational by defining it in terms of the proof obligations that recur in AlgoVeri—e.g., **global invariants, ghost state, inductive reasoning, and nontrivial loop specifications**—rather than relying on an intuitive label alone. Concretely, these include properties such as **reachability over an entire graph, balance conditions over a whole tree, or max-flow optimality**, where correctness depends on global relationships across the full structure rather than a single local step. Second, we will soften the phrasing around self-correction: rather than calling it an “emergent capability,” we will state the more direct empirical observation supported by Figure 4, namely that **effective iterative repair is concentrated in stronger frontier models and is much weaker in current open models**. Third, we will revise the statement about “generalizing from local logic to global properties” to make clear that this is an **interpretation of the category-level pattern** in Figure 3, rather than a directly isolated causal measurement. We will also revise Table 1 and related discussion to make benchmark comparisons more precise and easier to verify.
>
> >On sensitivity to language/tools.
>
> We agree that performance can shift across tool or library versions. However, AlgoVeri relies only lightly on external libraries, focuses on core verification patterns, and pins tool versions (including a fixed Mathlib commit), so we expect it to be less sensitive than benchmarks that depend heavily on evolving library lemmas or APIs.
>
> >On whether models could manipulate the LLM judge.
>
> In principle, yes. However, in our pipeline the judge is only a **secondary semantic filter after formal verification**, so such errors affect only the semantic-filtered metric. We discuss this in more detail in our response to Reviewer TJwi, including an initial audited sample on the newly added GPT-5.3-Codex results showing around 90% agreement with the intended semantic labels; the observed errors mainly came from out-of-scope judgments about proof correctness rather than failures on wrong-algorithm or shortcutting cases.
>
> >On presentation and limitations.
>
> We thank the reviewer for these suggestions. In revision, we will improve Table 2 readability, make Figure 5’s intended near-overlap result clearer, spell out abbreviations such as SMT at first use, and add a clearer limitations discussion.

---

> > ### Author Rebuttal · Reviewer_QDYe · 2026-04-01
> >
> > Thank you for the detailed responses.
> >
> > I am generally content with the authors' responses, but have one additional comment: a value of 0% on graph problems for powerful coding LLMs might mean that the problems are too difficult to be informative about model capabilities. This might reduces the value of the problem.
> >
> > Apart from that, I appreciate that the authors will revise their wording in some instances and improve the presentation, which will certainly improve the paper. In general, the main weakness, which is the size of the benchmark, remains.
> >
> > Overall, I think my assessment of the paper is fair and will keep my initial rating.

---

> > > ### Author Response · Authors · 2026-04-03
> > >
> > > Thank you for your thoughtful feedback and we sincerely appreciate your positive assessment.
> > >
> > > We agree that a benchmark category should ideally remain discriminative. At the same time, we believe very challenging subsets can still be highly informative when they expose a clear capability frontier rather than mere noise. In our case, the graph problems are not simply “harder at random”: they concentrate precisely the global reasoning burdens that motivate AlgoVeri, such as ghost state, reachability, and quantified optimality/correctness conditions. In this sense, even low current success rates are meaningful, because they identify an important unsolved region for future work rather than a saturated or uninformative part of the benchmark. We will make this point more explicit in the revision and continue expanding the benchmark so that it remains useful both for diagnosing current models and for tracking future progress.

---

### Official Review · Reviewer_oHS2 · 2026-03-11

**Soundness:** 3
**Presentation:** 3
**Significance:** 3
**Originality:** 3
**Overall Recommendation:** 5
**Confidence:** 4

**Summary:**

This paper presents ALGOVERI, a benchmark for evaluating verified code generation (vericoding) of classical algorithms across three dominant verification systems—Dafny, Verus, and Lean.
The work is well-motivated, with clear identification of the limitations of prior vericoding benchmarks (e.g., semantic misalignment, lack of global invariant reasoning, unfair cross-tool comparison), and ALGOVERI addresses these limitations with a carefully designed, expert-curated suite of 77 classical algorithm tasks with semantically aligned specifications across systems.

**Compliance With Llm Reviewing Policy:**

Affirmed.

**Key Questions For Authors:**

What are the properties in the verification conditions causing significant difficulties for LLMs to generate verified graph algorithms and advanced data structures?

**Limitations:**

yes

**Strengths And Weaknesses:**

# Strengths
- The paper offers strict semantic alignment of specifications across Dafny, Verus, and Lean isolates problem difficulty from toolchain effects, enabling a fair cross-paradigm comparison of neuro-symbolic reasoning for vericoding.
- The quality control measures—expert curation, formal well-formedness checks (local satisfiability, necessity against degeneracy), and a semantic validator to filter cheating/algorithmic degeneracy—ensure ALGOVERI is a rigorous, reliable benchmark for the community.
- The paper also provides valuable qualitative insights into the tradeoffs of different verification paradigms (SMT vs. ITP), making a compelling case for why multi-framework evaluation is necessary despite Dafny’s higher baseline performance.

# Weaknesses
The size of the bench is too small, with only 77 algorithms.

---

> ### Author Rebuttal · Authors · 2026-03-30
>
> We thank the reviewer for the positive assessment and for recognizing the value of strict semantic alignment and the qualitative insights on SMT vs. ITP tradeoffs.
>
> > Benchmark size.
>
> Because this concern was raised by multiple reviewers, we provide this more detailed clarification here and refer to it in our other responses where space is limited.
>
> We agree that 77 problems are modest in absolute terms. However, this is a deliberate design choice: AlgoVeri contains **231 expert-curated verification tasks** across Dafny, Verus, and Lean, with strict cross-language alignment. Our goal is not to maximize raw task count, but to build a benchmark where the same algorithmic problem can be compared fairly across fundamentally different verification paradigms. Each task is manually formalized and reviewed, and representative high-complexity tasks additionally undergo formal well-formedness checks to rule out vacuous or degenerate specifications. We believe this level of curation is important for benchmark reliability.
>
> Although modest in size, AlgoVeri is broad in algorithmic coverage: it spans complex data structures, sorting and order statistics, graph algorithms, dynamic programming/greedy methods, and mathematical algorithms. We also find that it is already sufficiently discriminative to separate both models and problem classes. For example, Gemini-3 Flash reaches 40.3% full correctness in Dafny, whereas GPT-OSS-120B reaches 13.5%, and the category-level breakdown shows a clear “complexity cliff,” with all models performing substantially worse on graph algorithms and advanced data structures than on simpler sorting and data-structure tasks. This suggests that the current benchmark already exposes meaningful capability differences.
>
> We nevertheless agree that scaling the benchmark is important. Importantly, we view this as **future-facing rather than corrective**: the current benchmark is already challenging for existing models, especially in graph algorithms and advanced data structures, but expanding coverage will help ensure that AlgoVeri remains informative as model capabilities improve. In particular, we are extending AlgoVeri in two directions: (i) adding more advanced data structures and algorithms, such as B-trees and KD-trees, to better reflect realistic verification challenges; and (ii) adding more compositional problems, including olympiad-style tasks that combine multiple algorithmic components, such as dynamic programming on trees or greedy reasoning over topological orderings. Such compositions of basic algorithms and data structures are also ubiquitous in real software and systems, where correctness often depends on how multiple verified components interact rather than on each component in isolation.
>
> > Properties causing difficulty for graph algorithms and advanced data structures.
>
> Our results suggest that the main challenge is not just implementation complexity, but the need to prove global structural properties rather than local functional behavior. Several proof obligations repeatedly appear in the hardest categories:
> + Ghost state / auxiliary abstractions. Many graph algorithms require proof-only objects, such as visited sets, discovery orders, or flow maps, which have no direct runtime counterpart.
> + Global inductive invariants. Advanced data structures such as red-black trees require showing that local updates preserve a global invariant over the entire structure..
> + Frame and heap reasoning. When updating one component of a graph or heap structure, the proof must also establish that unrelated parts remain unchanged; models often under-specify these obligations.
> + Termination arguments. Recursive traversals and graph search procedures require nontrivial decreasing measures, such as shrinking unexplored regions or structurally decreasing recursion.
> + Quantifier-heavy optimality/correctness conditions. Graph problems often involve nested quantified properties, such as reachability, conservation, or optimality, which are difficult both for LLM generation and for downstream SMT reasoning.
>
> These difficulties compound in tasks such as Edmonds–Karp max flow, where the specification already requires reasoning about flow conservation, capacity constraints, and existential witnesses for global optimality. We will add a short discussion in the revision to make these verification barriers more explicit.

---

> > ### Author Rebuttal · Reviewer_oHS2 · 2026-04-01
> >
> > Thanks for the review. I keep my score to be positive.

---

### Official Review · Reviewer_rK7x · 2026-03-12

**Soundness:** 3
**Presentation:** 3
**Significance:** 4
**Originality:** 3
**Overall Recommendation:** 5
**Confidence:** 2

**Summary:**

This paper introduces ALGOVERI, a benchmark of 77 algorithmic tasks for evaluating LLMs on verified code generation across Dafny, Verus, and Lean. Experiments show that vericoding complex algorithms remains challenging. The authors also analyze test-time compute scaling and failure modes, showing that frontier models (like Gimini-3) benefit from iterative repair while open models (like GPT-OSS)  saturate early.  Besides, by an error analysis, authors further disclose that different language design will steer models towards different refinement goals.

**Compliance With Llm Reviewing Policy:**

Affirmed.

**Final Justification:**

This paper is well written, and the authors have addressed my previous concerns. I will maintain my initial score. My only remaining concern is the dataset size; if accepted, I encourage the authors to more clearly explain why this dataset size is reasonable for a broader audience.

**Key Questions For Authors:**

1. The benchmark currently contains 77 tasks. Do the authors have plans to expand the dataset, and how representative are these tasks of real-world verification workloads?

2. The study mainly analyzes inference-time repair dynamics. Do the authors expect that specialized training (e.g., RL or proof-aware pretraining) could significantly improve performance on ALGOVERI?

3. The constructed benchmark focus on texbook algorithm. How well does the benchmark reflect real-world verification tasks, such as verifying complex software systems?

**Limitations:**

yes

**Strengths And Weaknesses:**

## Strengths

- The paper addresses a challenging and increasingly relevant problem of verified code generation, which is critical for improving the reliability of AI-generated software.

- ALGOVERI provides a more fair comparisons across different verification paradigms (Dafny, Verus, and Lean). This is a meaningful contribution compared to prior benchmarks that focus on a single language.

- Detailed experimental analysis about the test-time compute dynamics (frontier models vs open-sources models)  and the affects of different language design provides useful insights into current limitations of LLM-based vericoding.

## Weakness

- The benchmark contains 77 problems, which is relatively small compared to other modern evaluation datasets. It is unclear whether this size is sufficient to draw conclusions about model capabilities.

- The specifications are curated manually by experts. While this ensures quality, it raises concerns about scalability and reproducibility.

- The use of an LLM judge to check algorithmic fidelity introduces an additional source of uncertainty and potential bias in evaluation.

---

> ### Author Rebuttal · Authors · 2026-03-30
>
> We thank Reviewer rK7x for the positive assessment and thoughtful questions. We address each point below. For brevity, on recurring issues such as benchmark size and the semantic validator, we give a short response here and provide more detailed discussion in our responses to Reviewers oHS2 and TJwi.
>
> > Q1: Plans to expand the dataset? How representative are these tasks?
>
> Yes. We are already extending AlgoVeri by adding (i) more advanced data structures and algorithms, such as B-trees and KD-trees, and (ii) more compositional problems, such as dynamic programming on trees or greedy reasoning over topological orderings. We view this as **future-facing rather than corrective**: the current benchmark is already challenging for existing models, but broader coverage will help it remain informative as stronger models emerge. The current 77 problems already span six CLRS-aligned categories (Figure 2) and cover the main proof obligations we want to study, including loop invariants, structural invariants, ghost state, inductive reasoning, frame reasoning, and quantified correctness conditions. They are also already diagnostically useful: the current suite clearly separates both model families and algorithmic categories. We discuss the benchmark-size rationale in more detail in our response to Reviewer oHS2.
>
> > Q2: Could specialized training improve performance?
>
> Yes, we believe specialized training is likely to improve performance on this benchmark. Our error analysis suggests that the bottleneck is often not merely understanding the high-level proof idea, but expressing that reasoning in the verifier-specific formal artifacts required by each system—for example, loop invariants, ghost state, auxiliary lemmas, or tactic sequences. This is closely analogous to formal math theorem proving in Lean, where a model may capture the rough mathematical argument but still struggle to produce a correct formal proof script; in that setting, specialized training has already led to substantial gains. For this reason, verifier-specific data, fine-tuning, reinforcement learning, or agentic repair methods could plausibly yield substantial improvements here as well. We see AlgoVeri as a useful controlled testbed for measuring such progress.
>
> > Q3: How well does the benchmark reflect real-world verification tasks?
>
> AlgoVeri targets the foundational reasoning patterns that many larger verified systems are built on, including ghost state, global invariants, inductive reasoning, frame reasoning, and quantified correctness conditions. In this sense, AlgoVeri is closely connected to real-world software verification, even though it does not aim to provide a full end-to-end systems evaluation. This connection will become even stronger in our planned expansion, e.g., with B-trees and KD-trees.
>
> > On the semantic validator.
>
> We would like to clarify that the LLM judge is only a **secondary semantic filter** applied after formal verification, to catch issues such as wrong-algorithm implementations, disallowed library usage, or similar shortcutting behaviors. In an initial rebuttal audit on the newly added GPT-5.3-Codex results, agreement with the intended semantic labels was around **90%**; the few observed errors mainly came from out-of-scope judgments about proof correctness. We provide a more detailed discussion in our response to Reviewer TJwi.

---

> > ### Author Rebuttal · Reviewer_rK7x · 2026-04-03
> >
> > The concerns regarding the benchmark’s size and coverage remain. Overall, I believe my assessment of the paper is fair, and I will maintain my initial rating.

---

### Official Review · Reviewer_TJwi · 2026-03-12

**Soundness:** 3
**Presentation:** 3
**Significance:** 3
**Originality:** 3
**Overall Recommendation:** 4
**Confidence:** 2

**Summary:**

Vericoding is a programming paradigm where an AI model generates code along with a formal specification (the rules) and a machine-checkable proof (the mathematical evidence) that the code is correct. The ALGOVERI benchmark evaluates this capability across 77 classical algorithms using three languages: Dafny, Verus  and Lean.

**Compliance With Llm Reviewing Policy:**

Affirmed.

**Final Justification:**

The reviewers addressed my concerns, I think my score is still fair.

**Key Questions For Authors:**

1.  **On Specification Alignment:** Since you enforced identical contracts across all languages, and acknowledge the limitations related to this, to what extent do you think the performance collapse in Lean is due to **non-idiomatic specifications** versus the model's actual inability to construct proofs? Do you think it would be achievable to have aligned, AND idiomatic specifications?
2.  **On the Semantic Validator:** How did you verify the accuracy of the **LLM Judge** itself? Was there a human-in-the-loop audit to ensure it didn't accidentally fail valid but highly creative solutions?
3. **On the Semantic Validator's Metrics:** "Given that the LLM Judge serves as the final arbiter for semantic correctness, can you provide standard classification metrics (like accuracy, precision, or F1-score) for its performance?

**Limitations:**

Yes.

**Strengths And Weaknesses:**

* **Strengths:**
    * **Strict Parallel Alignment:** Enforces identical functional "contracts" across all three languages, ensuring models are tested on the exact same logical difficulty regardless of the tool.
    * **Algorithmic Rigor:** Targets "textbook" algorithms that require reasoning about **global properties** (like tree balance or graph reachability) rather than simple local operations.
    * **Anti-Cheating Safeguards:** Uses an **LLM Judge** and formal audits to ensure models don't pass with "vacuous success" (e.g., writing a trivial function that technically satisfies a weak proof).
    * **Expert Curation:** Specifications are manually authored and reviewed by formal methods experts rather than being automatically generated, ensuring high quality.

* **Weaknesses:**
    * **Translation Hardness:** Maintaining strict alignment can result in **non-idiomatic specifications**, particularly in Lean, which may add a "hidden" difficulty layer for models.
    * **Limited Scope:** While rigorous, the dataset contains only **77 problems**. This is far smaller than some unaligned benchmarks that aggregate thousands of simpler tasks.
    * **Unquantified Judge Reliability:** The benchmark relies heavily on an LLM Judge to serve as a "semantic filter" (ensuring models don't cheat the specifications). However, the paper does not report standard classification metrics (such as accuracy, precision, recall, or F1-score) for this judge, leaving its exact false-positive and false-negative rates unknown.

---

> ### Author Rebuttal · Authors · 2026-03-30
>
> We thank Reviewer TJwi for recognizing the strict parallel alignment, algorithmic rigor, and anti-cheating safeguards of AlgoVeri.
>
> > On specification alignment / non-idiomatic Lean specifications.
>
> We agree that strict alignment can introduce some non-idiomaticity, especially in Lean, whose natural style often favors inductive definitions and theorem statements that are not the same as those used in SMT-based systems. We therefore do not want to claim that this effect is zero. However, our error analysis suggests that it is unlikely to be the main driver of the performance collapse. In Figure 6(c), Lean failures are dominated by **hallucination/search errors** (e.g., citing non-existent lemmas or tactics) and **verification errors** (incorrect proof reasoning), rather than by low-level syntax failures. This suggests that the main bottleneck is the difficulty of proof search and proof construction in Lean, not merely the surface form of the aligned specification. More generally, we believe “strictly aligned” and “fully idiomatic in every system” are goals that are only partially compatible: Lean naturally favors constructive definitions and lemma hierarchies, whereas Dafny and Verus are built around SMT-friendly contracts, loop invariants, and ghost state. In this work, we prioritized alignment because it is necessary to separate toolchain effects from problem difficulty in a cross-paradigm benchmark. We will clarify this tradeoff more explicitly in the revision.
>
> > On the semantic validator and its reliability.
>
> We would like to clarify that the LLM judge is used only as a **secondary semantic filter** in an otherwise fully automatic pipeline. A solution must first pass formal verification; only then is the judge applied to detect a narrow class of failures that the functional specification may not exclude, such as cheating, disallowed library usage, or implementing a different algorithm than the one requested in the prompt. Thus, the judge is not intended to assess proof correctness; proof validity is already determined by the verifier.
>
> We agree that the paper should quantify this component more explicitly. In our rebuttal analysis, we examined judge decisions on the newly added GPT-5.3-Codex results, which provide a fresh set of semantic-filtering cases. On this annotated sample, agreement between the LLM judge and the intended semantic labels is around **90%**. Importantly, the few observed errors mainly arise when the judge overreaches and comments on proof correctness, which is outside its intended role. At the same time, in this initial audited sample, the judge correctly identified all observed cases of wrong-algorithm implementations, disallowed library usage, and similar shortcutting behaviors.
>
> Regarding “creative” solutions, our intended policy is to distinguish **algorithmic identity** from **implementation style**: a functionally correct but algorithmically different solution should be rejected by design, whereas a nonstandard or optimized implementation of the intended algorithm should be accepted. In principle, an LLM judge could still wrongly reject a highly creative implementation that preserves the intended algorithmic identity, but we did not observe such cases in this initial audit.
>
> There is **no human-in-the-loop in the benchmark evaluation pipeline itself**: AlgoVeri remains fully automatic, and this annotated audit is only for assessing the reliability of the judge, not for determining benchmark outcomes.
>
> >On benchmark size.
>
> We agree that the benchmark is modest in absolute size; we provide a more detailed discussion of this design tradeoff and expansion plan in our response to Reviewer oHS2.

---

> > ### Author Rebuttal · Reviewer_TJwi · 2026-04-01
> >
> > Thank you for the rebuttal.

---

### Decision · Program_Chairs · 2026-04-30

**Decision:**

Accept (spotlight)

**Comment:**

This is a clear accept -- All the reviewers unanimously agreed on the merits of the paper.